# Gender imbalance amongst promotion and leadership in academic surgical programs in Canada: A cross-sectional Investigation

Jennifer Hunter[1][ⓘ], Helen Crofts[1‡], Alysha Keehn[2‡], Sofie Schlagintweit[3‡], Jessica G. Y. Luc[4‡], Kelly A. Lefaivre[1][ⓘ]*

1 Department of Orthopaedics, University of British Columbia, Vancouver, British Columbia, Canada, 2 Division of General Surgery, University of Calgary, Calgary, Alberta, Canada, 3 Division of Plastic Surgery, University of British Columbia, Vancouver, British Columbia, Canada, 4 Division of Cardiovascular Surgery, University of British Columbia, Vancouver, British Columbia, Canada

ⓘ These authors contributed equally to this work.
‡ These authors also contributed equally to this work.
* kelly.lefaivre@vch.ca

**Data Availability Statement:** All relevant data are within the manuscript.

## Abstract

### Background

Women are underrepresented at higher levels of promotion or leadership despite the increasing number of women physicians. In surgery, this has been compounded by historical underrepresentation. With a nation-wide focus on the importance of diversity, our aim was to provide a current snapshot of gender representation in Canadian universities.

### Methods

This cross-sectional online website review assessed the current faculty listings for 17 university-affiliated academic surgical training departments across Canada in the 2019/2020 academic year. Gender diversity of academic surgical faculty was assessed across surgical disciplines. Additionally, gender diversity in career advancement, as described by published leadership roles, promotion and faculty appointment, was analyzed.

### Results

Women surgeons are underrepresented across Canadian surgical specialties (totals: 2,689 men versus 531 women). There are significant differences in the gender representation of surgeons between specialties and between universities, regardless of specialty. Women surgeons had a much lower likelihood of being at the highest levels of promotion (OR: 0.269, 95% CI: 0.179–0.405). Men surgeons were statistically more likely to hold academic leadership positions than women (p = 0.0002). Women surgeons had a much lower likelihood of being at the highest levels of leadership (OR: 0.372, 95% CI: 0.216–0.641).

### Discussion

This study demonstrates that women surgeons are significantly underrepresented at the highest levels of academic promotion and leadership in Canada. Our findings allow for a

**Funding:** The authors received no specific funding for this work.

**Competing interests:** The authors have declared that no competing interests exist.

direct comparison between Canadian surgical subspecialties and universities. Individual institutions can use these data to critically appraise diversity policies already in place, assess their workforce and apply a metric from which change can be measured.

## Introduction

The importance of the social discussion and scientific study of diversity has never been as prevalent as it is today. In Canada, our government demonstrates the importance diversity has to Canadians by promoting policies that eliminate workplace discrimination and gender inequity at the provincial and federal levels [1–4]. Evidence from the corporate world shows that increasing diversity improves organizational success and the financial bottom line across disciplines [5–8]. If industries continue to advance gender equity, studies predict improved outcomes could add an additional 4–9% to the Canadian GDP [5].

Medical organizations have placed an increasing importance in diversity. A simple Google search yields a plethora of diversity statements and policies from provincial and national medical organizations, individual hospitals, regional health authorities, and the Public Health Agency of Canada. They recognize that gender diversity research shows patients treated by women physicians had lower rates of mortality, and women surgeons had decreased rates of 30-day post-operative mortality; and know that, as in the business world, increasing gender diversity within their organizations would benefit their patients [9, 10]. However, the metrics on the success of implemented diversity statements and policies are only just emerging and unclear in the literature [11–14].

Increasingly, academic institutions, and their affiliated medical faculties, are focusing on this topic. Each Canadian university has a diversity statement and/or office to represent ongoing engagement. However, the expanding wealth of international research in gender diversity within academic medicine reveals that women are underrepresented around the world in academic faculty positions and leadership roles [15–18]. For women scientists, they must also contend with institutional systemic gender biases to get access to grants, have their work published and then receive credit for their published work compared to their men peers [19–21]. Important Canadian funding agencies, including the Canadian Institute for Health Information and Canadian Institutes of Health Research, have published 3 independent studies in 2018 and 2019 citing gender bias against women grant applicants, grant peer-review and personnel awards [22–24].

When it comes to clinical academic medicine, surgical specialties present their own historic gender diversity challenges. While surgery used to be traditionally men, Canadian surgeons have seen the proportion of women within their ranks increase to 30%, and women trainees have increased from 32% to 43% of surgical residents in the last decade [25, 26]. There has been increasing interest in research related to gender inequities that exist in surgical careers and training programs [27–31]. Many surgical subspecialties have contributed to this expanding literature related to the gender disparities faced by trainees, surgical faculty, the lack of women in leadership positions, and represented in academic authorship or on editorial boards in their respective fields [32–57]. However, comparisons between subspecialties and different university academic or training environments are lacking. There is very limited data examining the gender distribution and achievements of Canadian surgeons. Of the existing Canadian research, it has been shown that women surgeons are less likely to ascend to higher ranks of academic faculty or to leadership positions, and they make statistically significantly less money than their men colleagues [41, 58, 59].

With these ongoing challenges in mind, our primary aim was to evaluate gender diversity within all Canadian surgical academic departments, and our secondary aim was to compare differences across specialties and between universities.

## Methods

Publicly available website sources were used to collect information on surgical faculty for the 17 medical schools in Canada in the 2019/2020 academic year. Data extraction was checked in duplicate by other author(s) in a random sample of fields across the 17 medical schools. Specialties were included when they were listed as a division under the department of surgery in 10 or more out of the 17 medical schools. The following nine specialties met this requirement (≥10/17): General Surgery, Vascular Surgery, Cardiac Surgery, Plastic Surgery, Otolaryngology, Thoracic Surgery, Orthopaedic Surgery, Urological Surgery, and Neurosurgery. Other specialties that did not meet the threshold of membership (i.e. listed as a division under the department of surgery in <10/17 medical schools) and hence were not included were specialties such as: Ophthalmology, Obstetrics and Gynecology, and Interventional Radiology, among others with variability by site.

Each faculty member was listed according to the departmental or divisional website, and the variables recorded were gender, specialty, university appointment, and additional education (Masters, PhD, and other). Provincial college listings were used to cross reference gender when needed.

Leadership positions were collected as reported freeform, with an unlimited number recorded per faculty member. These were later subcategorized as follows: Department Head—University, Associate Department Head—University, Division Head—University, Section Head (subsection. of Division)—University, Director Post-Graduate Medical Education, Director Continuing Medical Education, Director Under-Graduate Medical Education, Hospital Chief of Surgery, Hospital Chief of Staff or Medical Director, Leader of Hospital Department/Division, and other University, Research or Hospital Leadership role. For analysis, these were subcategorized as Higher levels of Leadership (Department Head, Division Head, Section Head, Associate/Vice Head), and Educational Roles (Director Post-Graduate Medical Education, Director Continuing Medical Education, Director Under-Graduate Medical Education).

Resident information was procured from the residency programs for each specialty; it was made available with variable specificity given differential privacy concerns. In all cases, specialty, school, and gender were available.

All statistics were performed by a graduate level trained statistical consultant. All analyses were performed using R (R Core Team, 2019) including univariate analysis and regression modeling and prediction construction [60]. Summary statistics are reported when needed. Differences in proportions were tested using a two-tailed Z-test. Chi-squared tests were used to test independence between two categorical variables. For the regression modelling, generalized linear models were used with appropriate link functions depending on the type of response variable. Logistic regression was used when the outcome was binary, and ordinal logistic regression was used for ordered categorical responses. Adjusted odds ratios are reported for logistic and ordinal logistic regression models and appropriate confidence intervals are given. All tests were performed at the p = 0.05 significance level.

## Results

There were 3,220 academic surgical faculty members affiliated with nine surgical specialties and distributed across all 17 medical schools in Canada. There were 2,689 men (0.84) and 531

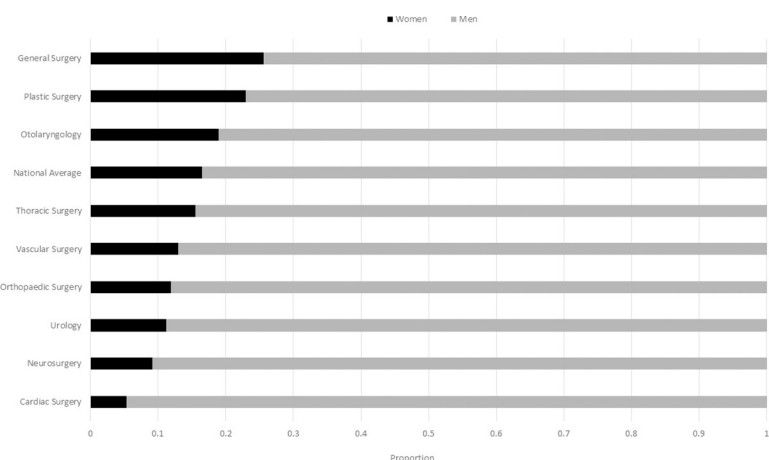

**Fig 1. Proportion of women to men academic surgeons by specialty.** Ordered from highest proportion of women surgeons to lowest proportion. Significant differences exist between specialties (p<0.0001).

women (0.16) (S1 Table). The gender distribution by specialty ranged from a high of 0.26 women (General Surgery) to a low of 0.05 (Cardiac Surgery) (p<0.0001) (Fig 1).

The gender distribution by university ranged from a high of 0.24 women at the Université de Sherbrooke, to a low of 0.12 at the University of British Columbia (p = 0.05) (Fig 2). A higher proportion of women (0.29) than men (0.23) surgical faculty hold an additional higher university degree (Masters and or PhD) (p = 0.0023).

## Academic appointment/promotion

Reported academic faculty promotion levels were assessed. Some departments classified their faculty into either 'clinical' or' academic' streams, while others did not have formal distinctions classifying their entire faculty into a single stream. Univariate comparison between gender of surgical faculty and level of promotion showed a strong statistical significance (p<0.001) (Fig 3). Women faculty are most frequently seen in the lowest levels of promotion (none, Lecturer, Clinical Instructor, Clinical Assistant Professor, Assistant Professor). Among all listed faculty,

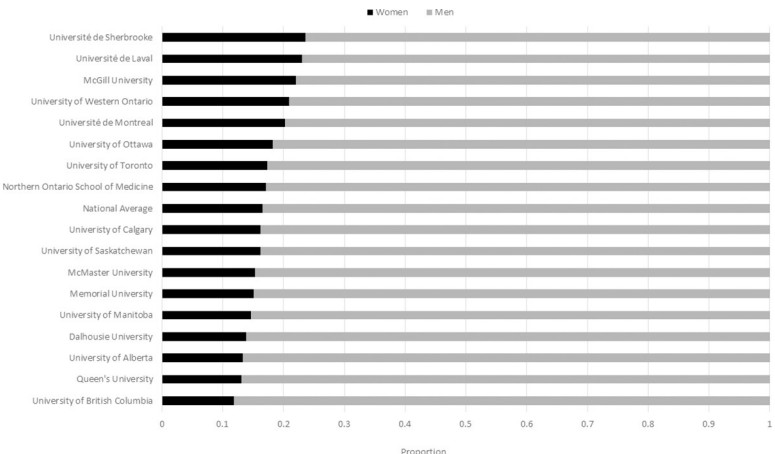

**Fig 2. Proportion of women to men academic surgeons by university.** Ordered from highest proportion of women surgeons to lowest proportion. Significant differences exist between universities (p<0.05).

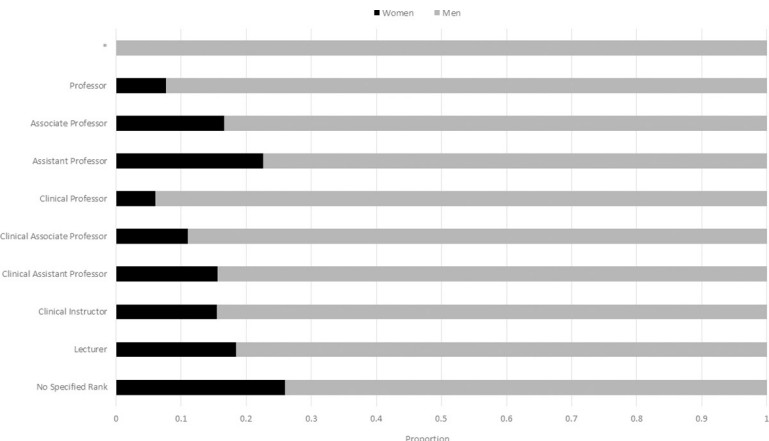

**Fig 3. Proportion of women to men surgical faculty by level of academic promotion.** Univariate comparison between gender of surgical faculty and level of promotion showed a strong statistical significance (p<0.001). * = Combined category of Professor Emeritus/Honorary Associate Professor/Distinguished Professor.

women were nearly twice as likely (0.11) than men (0.06) to not have a specified faculty rank even though they were included in faculty lists. In academic promotion, women faculty are relatively clustered at Assistant Professor, accounting for (0.36) of the entire women faculty nationally, and the highest relative representation to men colleagues (Fig 3). As a proportion of their own gender, men and women are equally likely to be at the academic rank of Associate Professor (0.16); however, only 0.17 of Associate Professors overall are women (Fig 3).

Women are least represented at the highest levels of promotion, Clinical Professor (0.017) and Professor (0.055), and absent from the Professor Emeritus rank (p<0.001). Of the 376 professors of surgery across Canada, 29 are women surgeons (0.01 of all faculty) compared with 347 men surgeons (0.11 of all faculty) (p<0.001). Regression modeling to account for the impact of university, specialty and higher degrees and gender on attaining the highest levels of promotion (Professor, Professor Emeritus) showed women had a much lower likelihood of being at the highest level of promotion (OR: 0.269, 95% CI: 0.179–0.405). Similarly, when considering promotion on an overall 5 level categorical scale, and accounting for the same factors, women were less likely to be at a higher level of promotion (OR: 0.54, 95% CI: 0.456–0.661).

A prediction model to assess the impact of university affiliation in women attaining the highest level of promotion while controlling for specialty and higher degrees was run. The reference university was that with the highest univariate proportion of women at the highest level of promotion (University of Toronto, 0.15) (Fig 4).

This prediction analysis was also performed by specialty, with the reference specialty being Cardiac Surgery, as the specialty with the highest univariate representation of women at the highest level of promotion (0.22) (Fig 5).

## Leadership roles

Within the national surgical faculties, 762 surgeons were noted to hold at least one position of leadership. Men held 0.85 (n = 649) of these positions and women 0.15 (n = 113). The difference in proportion of men (0.32) and women (0.26) in some kind of leadership position was statistically significant (p = 0.002).

Women are more frequently found in education roles. The gender distribution of the residency program directors is 112 (0.042) men and 24 (0.047) women across the specialties (p = 0.815). Two specialties, Vascular Surgery and Cardiac Surgery, had no women in this

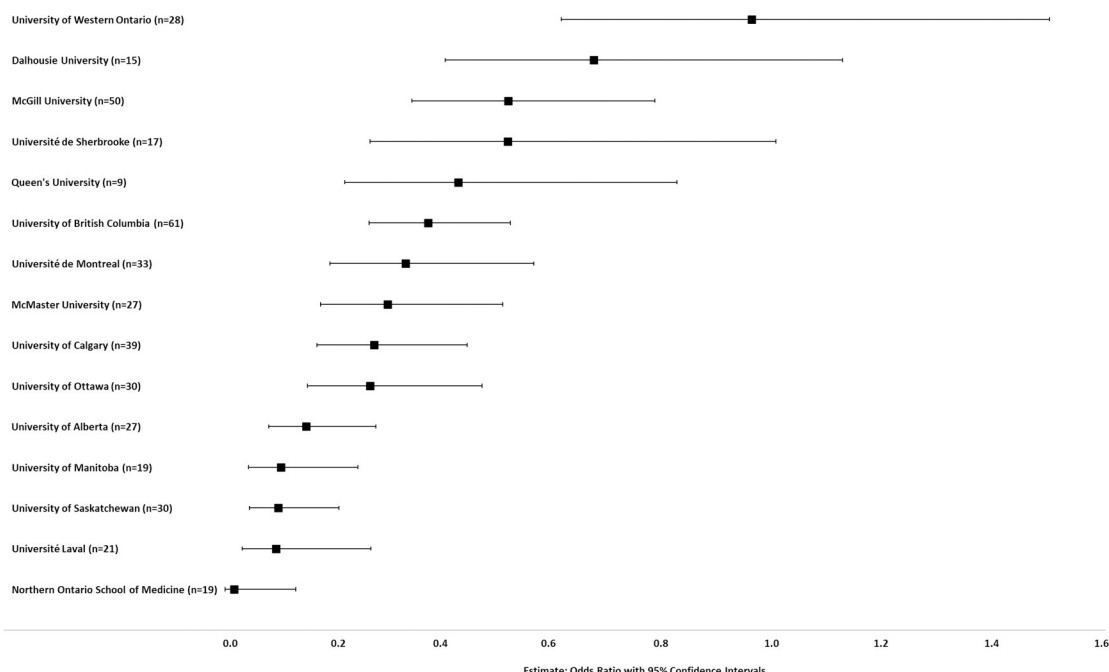

**Fig 4. Prediction modeling of the impact of university affiliation on likelihood of women academic surgeons attaining highest levels of promotion (Professor, Professor Emeritus/Honorary Associate Professor/Distinguished Professor).** Controlled for specialty and higher degrees. Odds ratio with 95% confidence interval graphed for each university compared to the reference university (University of Toronto, 0.15, highest proportion of women at the highest levels of promotion).

educational leadership position. Although the role is less reliably reported, of those that reported undergraduate education directors they are disproportionately women, with 0.23 being women (p = 0.04).

Within the institutional academic leadership positions of Division Head/Chair or Department Head/Chair, men were statistically more likely to hold these positions than women

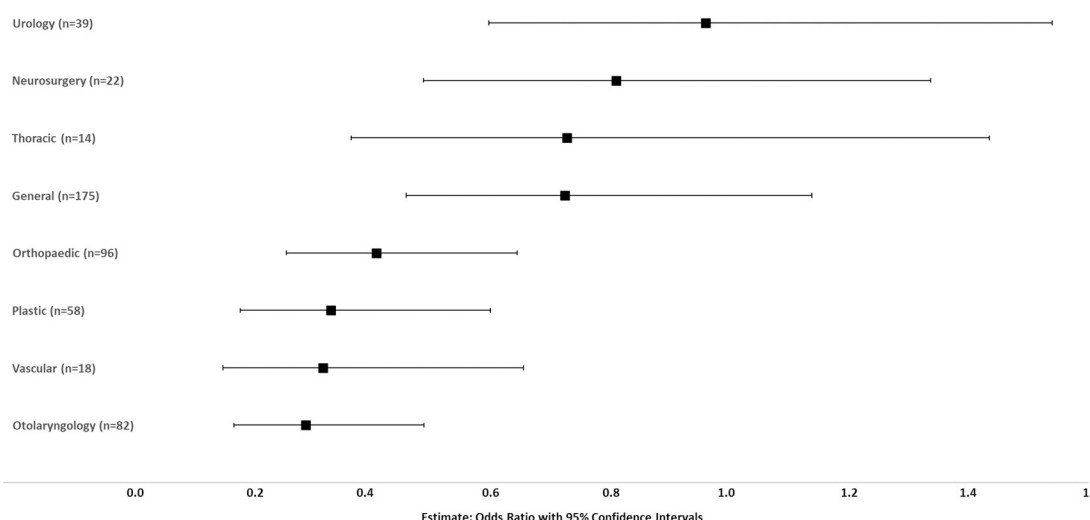

**Fig 5. Prediction modeling of the impact of specialty on likelihood of women academic surgeons attaining highest levels of promotion (Professor, Professor Emeritus/Honorary Associate Professor/Distinguished Professor).** Controlled for university and higher degrees. Odds ratio with 95% confidence interval graphed for each specialty compared to the reference specialty (Cardiac Surgery, 0.22, highest proportion of women at the highest levels of promotion).

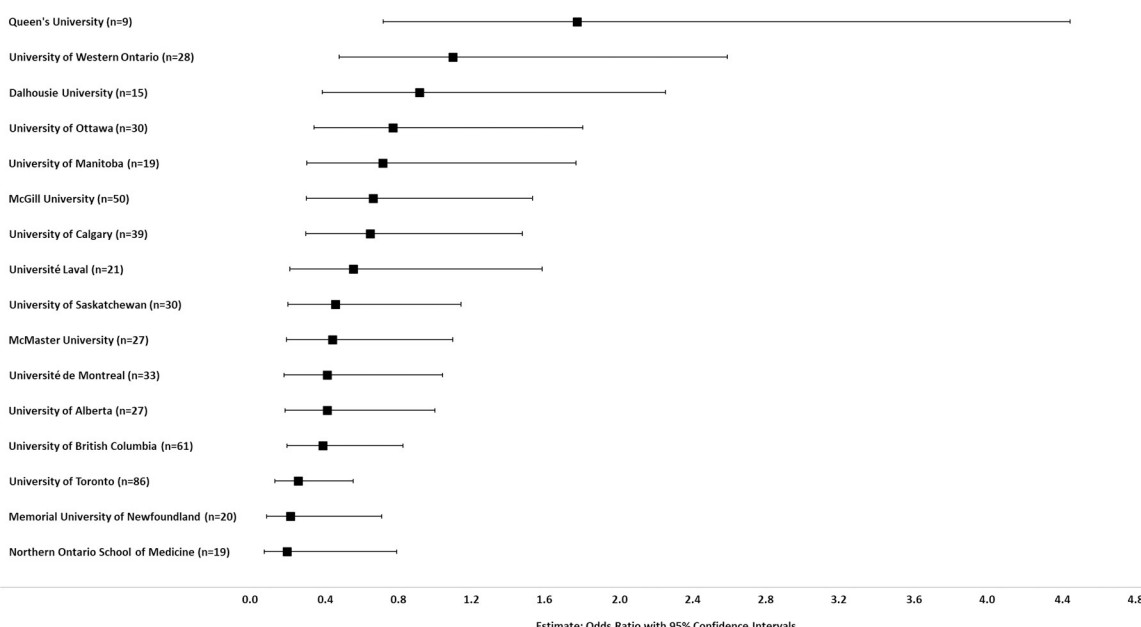

**Fig 6. Prediction modeling of the impact of university affiliation on likelihood of women academic surgeons attaining highest levels of leadership (Department, Division or Section Head).** Controlled for specialty and higher degrees. Odds ratio with 95% confidence interval graphed for each university compared to the reference university (Université de Sherbrooke, 0.17, highest proportion of women at the highest levels of leadership).

(p = 0.0002), with 157 men and only 13 women leading their peers across the country. Specifically, of 27 departments reporting a head, there are only 2 women sitting as a Department Head across the country.

Regression modeling to account for the impact of university, specialty and higher degrees and gender on attaining the higher levels of Leadership (Department Head, Division Head, Section Head, Associate/Vice Head) showed women had a much lower likelihood of being in higher levels of leadership (OR: 0.372, 95% CI: 0.216–0.641).

Prediction modeling was again used to assess the impact of university affiliation in women attaining the highest level of leadership, while controlling for specialty and higher degrees. The reference university was that with the highest univariate proportion of women at the highest level of promotion (Université de Sherbrooke, 0.17) (Fig 6).

A similar leadership analysis was performed by specialty, with the reference specialty being Thoracic Surgery, as the specialty with the highest univariate representation of women at the highest levels of leadership (0.14) (Fig 7).

## Canadian surgical resident cohort

There were 1694 surgical trainees enrolled across eight surgical specialties within Canada in the 2019/2020 academic year. The gender breakdown of this group was 1,060 men surgical trainees (0.63) and 634 women surgical trainees (0.37) (S2 Table). There is a significant difference in the gender breakdown of residents across the differing specialties (p<0.00001). The proportion ranged from a high of 0.53 women (General Surgery) to a low of 0.20 (Cardiac Surgery) (Fig 8).

There was also a significant difference between universities in the proportion of women to men residents (p = 0.0029). The proportion of residents by university revealed the highest proportion of women residents at Memorial University (0.64) and the lowest proportion at McGill University (0.31) (Fig 9).

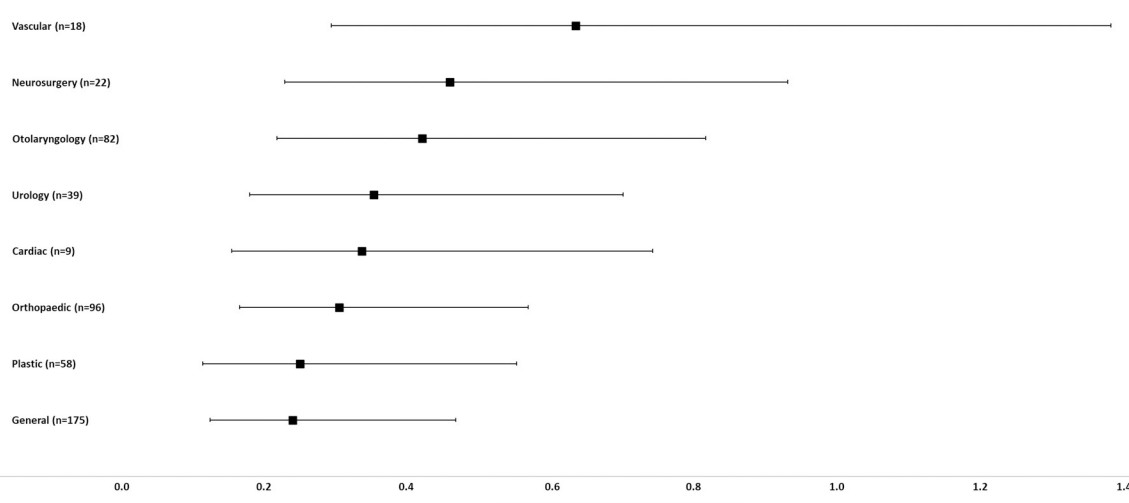

**Fig 7. Prediction modeling of the impact of surgical specialty on likelihood of women academic surgeons attaining highest levels of leadership (Department, Division or Section Head).** Controlled for university and higher degrees. Odds ratio with 95% confidence interval graphed for each specialty compared to the reference specialty (Thoracic Surgery, 0.14, highest proportion of women at the highest levels of leadership).

Regression modeling demonstrated that the number of women surgeons in higher leadership roles did not influence the number of women residents at a given university (p = 0.9194). However, regression modeling showed that the more women in educational roles for a specialty results in a larger proportion of women residents in that specialty (p = 0.046).

## Discussion

To borrow from the words of Nonet Sykes: "equity is the provision of opportunity, networks and resources that support each of us to reach our full potential." Diversity and inclusion, while not the sole components of equity in the workplace or in medicine, play important roles in having the opportunity for equity. Diversity among those who work in the medical field matters to our patients. This is supported by evidence showing that diversity can produce improved health outcomes for patients and that a more diverse health-care workforce can help reduce racial and ethnic disparities in patient care [61]. Gender is just one component of

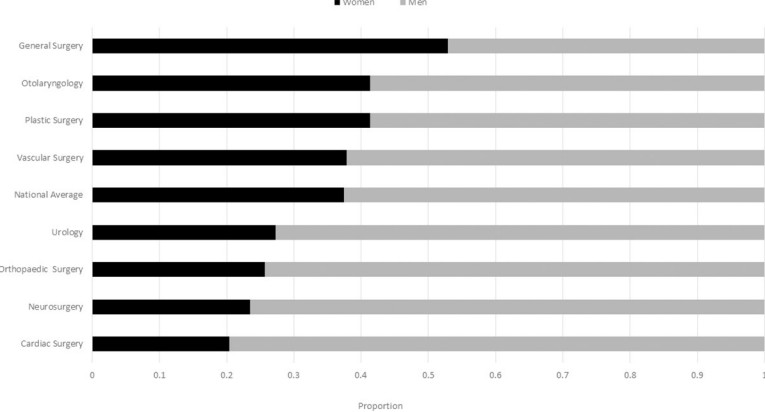

**Fig 8. Proportion of women to men surgical residents by specialty.** Ordered from highest proportion of women surgeons to lowest proportion. Significant differences exist between specialties (p<0.00001).

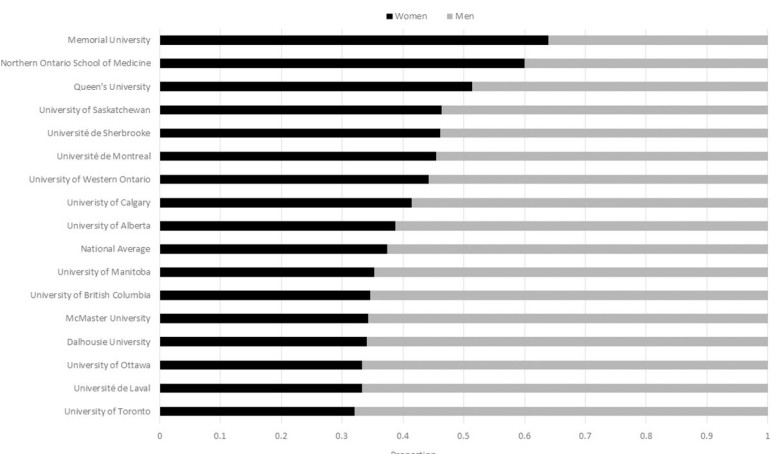

**Fig 9. Proportion of women to men surgical residents by university.** Ordered from highest proportion of women surgeons to lowest proportion. Significant differences exist between universities (p<0.01).

diversity in healthcare that has been increasingly studied, and highlighted in importance, in recent years.

Continued lack of diversity within the academic faculty leads to a persistent opportunity gap between the genders. Increases in the number of women surgeons overall have not translated directly to increased representation at the highest academic promotion or leadership levels, with a study showing no narrowing of the gap over the last 35 years [62]. It is recognized that a certain representation is needed to see 'shift', and in surgery in Canada we fall below the "critical mass" of 30% representation at higher levels of leadership and academia which would lead to culture change [63].

This study joins others among surgical subspecialties in highlighting ongoing gender disparity with regards to women surgeons within academia [32–37, 40, 41, 43–48, 51, 52, 54–58, 64]. While our investigation highlights the current Canadian environment, the themes of ongoing gender disparity are not isolated to this country; with specific studies noting that inequities exist in Plastic Surgery, Neurosurgery and General surgery across North America, the EU and Oceania [39, 45, 46, 65, 66]. Barriers to academic progression are multi-faceted and complex. Previous studies suggest that women are less likely to be invited to speak at grand rounds, they face significant bias when contending with their peers for grants or funding, are often underrepresented on editorial boards and are underrepresented and under cited within published literature of their field [19–24, 67, 68]. These factors are reflected in metrics that promotions committees use when assessing a surgeon's consideration for faculty promotion and leadership roles, and can put women at a disadvantaged position. Equally importantly, as advocacy and mentorship have been identified as crucial to career development, we note that women surgeons are less likely to have gender-specific mentorship given the lack of women at more advanced promotional rank as cited extensively above [69–72].

Barriers to gender equity also exist at the leadership level with the Canadian surgical specialties. Our study shows that men predominantly hold leadership positions amongst surgical faculty, and that this holds true across all specialties and universities, which mirrors findings of smaller specialty specific studies in the literature [64]. It also mirrors work published by our international colleagues showing that women are underrepresented in medical leadership in the EU, Australia/Oceania and all over North America [17, 18, 66]. Studies on leadership, promotion and performance evaluation have been widely published fields outside of medicine. Studies by leaders within the business world have identified that men will apply for a position

if they have satisfied 60% of the application criteria but women will limit their application until they check off 100% of the criteria; thus, contributing to women applying to 20% fewer jobs than men [73]. Studies on workplace biases also reveal that women are promoted or hired based on their previous experience whereas men are considered for their future potential making it difficult to break into leadership roles [74]. In the United States military, objective evaluations showed no difference in performance between genders but subjective evaluations showed significant gender biases in the language used in evaluations [75]. These implicit biases that exist in evaluating men versus women job applicants are difficult to mitigate as they are by definition unconscious. Strategies to help reduce implicit biases in hiring processes have been studied and are effective at increasing gender diversity. At an academic health care center, departments that received a brief self-assessment and educational session on implicit biases increased the hiring rate of women to twice that of departments that received no training [76]. In addition to individual bias training, larger initiatives that include university-based policy changes and leadership training may have a greater impact on addressing implicit biases [77].

Our study shows that the number of women surgical residents in Canadian training programs surpasses the number of academic staff by far, but that women remain greatly underrepresented in the highest levels of promotion and leadership. Advocacy and mentorship from early career stages will serve to bolster our community longitudinally [69–72]. Within our own networks, we must encourage women to take on leadership positions as studies show less than 10% of women are encouraged to apply for leadership positions by their superior [78]. At higher institutional levels, departments and universities alike need to formally consider what biases their current policies continue to propagate during hiring and promotion processes.

## Limitations

This cross-sectional study is limited by the quality, accuracy, and current standings of publicly available information on university websites. While more public facing roles such as department head or residency program director are consistently and reproducibly found on university-affiliated websites, the same cannot be said for every other academic or leadership role across all the specialties or schools and is at the discretion of each faculty. However, this represents an accurate assessment of the public reporting of leadership and promotion- which represents the forward-facing image of each organization. Also, these data provide a snapshot of the currently publicly listed Canadian surgical environment and cannot answer the questions of how this has changed over time. While the dataset is large in its entirety, at the highest levels of leadership and promotion, the absolute number of women remains low. In some instances, only a small n was available for the regression modelling and may impact the strength of the model. However, the models themselves have shown results in keeping with the global findings of the study, supporting the results along with those published by our peers with similar work in this field. Lastly, this study is limited to investigating gender and thus misses other diversity sectors. There are surgeons who identify as members of other minority groups of race, sexual orientation, ethnicity or religious affiliation who are equally importantly at risk of bias and intersectional bias in their career.

## Conclusion

Gender diversity within the Canadian surgical specialties continues to be an evolving issue. This study demonstrates that overall, women surgeons are underrepresented at the highest levels of academic promotion and leadership. Our data also allows for a direct comparison between surgical subspecialties and universities. Recognizing where a specialty or university stands compared to Canadian counterparts provides those in leadership positions the ability to

critically appraise diversity policies already in place, assess their workforce and have a metric from which change can be measured. Having women surgeons proportionally represented at every level of promotion and leadership can help instigate culture change and progress Canadian surgical departments toward true gender equity. In doing so, it will bring work environments into line with the diversity directives as espoused by collective institutional policies, governments and Canadian health agencies.

## Supporting information

**S1 Table. All academic surgical faculty across 17 medical schools in Canada.**
(XLSX)

**S2 Table. All resident surgeons across 17 medical schools in Canada.**
(XLSX)

## Acknowledgments

Our thanks to Dr. K. Aminotaljari, Dr. D.M. Roffey, and A. Wong for their assistance on this project.

## Author Contributions

**Conceptualization:** Jennifer Hunter, Helen Crofts, Alysha Keehn, Sofie Schlagintweit, Kelly A. Lefaivre.

**Data curation:** Jennifer Hunter, Helen Crofts, Alysha Keehn, Sofie Schlagintweit, Jessica G. Y. Luc.

**Formal analysis:** Jennifer Hunter, Helen Crofts, Alysha Keehn, Sofie Schlagintweit, Jessica G. Y. Luc, Kelly A. Lefaivre.

**Investigation:** Jennifer Hunter, Helen Crofts, Alysha Keehn, Sofie Schlagintweit, Jessica G. Y. Luc, Kelly A. Lefaivre.

**Methodology:** Jennifer Hunter, Helen Crofts, Alysha Keehn, Sofie Schlagintweit, Jessica G. Y. Luc, Kelly A. Lefaivre.

**Project administration:** Jennifer Hunter, Kelly A. Lefaivre.

**Supervision:** Kelly A. Lefaivre.

**Validation:** Jennifer Hunter, Helen Crofts, Alysha Keehn, Sofie Schlagintweit, Jessica G. Y. Luc, Kelly A. Lefaivre.

**Visualization:** Jennifer Hunter, Helen Crofts, Alysha Keehn, Sofie Schlagintweit, Jessica G. Y. Luc, Kelly A. Lefaivre.

**Writing – original draft:** Jennifer Hunter, Helen Crofts, Alysha Keehn, Sofie Schlagintweit, Jessica G. Y. Luc, Kelly A. Lefaivre.

**Writing – review & editing:** Jennifer Hunter, Helen Crofts, Alysha Keehn, Sofie Schlagintweit, Jessica G. Y. Luc, Kelly A. Lefaivre.

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
