## [Decision Letter · Decision Letter 0]

29 Jun 2021

PONE-D-21-18263

Gender imbalance amongst promotion and leadership in academic surgical programs in Canada: A cross-sectional review

PLOS ONE

Dear Dr. Lefaivre,

Thank you for submitting your manuscript to PLOS ONE. After careful consideration, we feel that it has merit but does not fully meet PLOS ONE’s publication criteria as it currently stands. Therefore, we invite you to submit a revised version of the manuscript that addresses the points raised during the review process.

Your study is interesting, the results as wells as comments/interpretations are relevant all over the world. I invited women only for review. Please follow their recommendations to improve the manuscript. From my side, I suggest to change the title: The term "review" my be misleading. In a review you describe what other authors have already published. Furtheermore, rreviewa are not very welcome in PLOS ONE. Your study, hosever, is a real investigations. I suggest to chage the title:.....a cross.sectional *investigation*.

We look forward to receiving your revised manuscript.

Kind regards,

Hans-Peter Simmen, M.D., Professor of Surgery

Academic Editor

PLOS ONE

Journal Requirements:

Reviewers' comments:

Reviewer's Responses to Questions

**Comments to the Author**

1. Is the manuscript technically sound, and do the data support the conclusions?

Reviewer #1: Partly

Reviewer #2: Yes

Reviewer #3: Yes

2. Has the statistical analysis been performed appropriately and rigorously? 

Reviewer #1: I Don't Know

Reviewer #2: Yes

Reviewer #3: Yes

3. Have the authors made all data underlying the findings in their manuscript fully available?

Reviewer #1: No

Reviewer #2: Yes

Reviewer #3: Yes

4. Is the manuscript presented in an intelligible fashion and written in standard English?

Reviewer #1: Yes

Reviewer #2: Yes

Reviewer #3: Yes

5. Review Comments to the Author

Reviewer #1: • This study describes a cross-sectional online website review of the current faculty of surgical departments in Canada with the focus on gender distribution. The aim was to analyse the proportion of female in promotion and leadership

• The data is put in relation to the politic of Canada, which tries to avoid gender disparity, as well as in context to the literature available to the topic and important networks like linkedIn.

• The data do support an underrepresentation of woman in surgery for promotion and leadership in general and relative to the number of trainees. The regression models may have been performed with very small numbers (n), leading to large confidence intervals. Therefore It would be important to show a flow chart of the numbers assessed (initially and for each surgical specialty) and the numbers excluded (e.g. Obstetrics and Gynecology) and to add n to the figures (especially fig 4 to 7). With small n the regression may not be adequate statistically.

• Although the findings of the study are not new, it is impressive how we still underuse the potential of diversity: diverse team get better results, which is essential in research and patient treatment. Moreover, we as a society cannot afford to train and develop females and not benefit from their knowledge. Therefore it is crucial to acknowledge how low the percentage of female is in position of promotion or leadership

• The study conforms to the STROBE guidelines

• The details are sufficient for the study to be reproduced

• The manuscript is well organized, but some improvement ist suggested for better understanding:

o Lines 300 to 303: make clear that women are 20% less likely to apply for a given job

o Label the x-axis in the figures 4-7, so it is self-explaning

o Report all n

o Show a flow chart and a chart with all the data according to PLOS ONE policy

o Some abbreviations are not explained, please elaborate

Reviewer #2: The authors performed a cross-sectional online website review of current faculty listings for 17 university-affiliated academic surgical training departments across Canada in the 2019/2020 academic year. They conclude that women surgeons are significantly underrepresented at the highest levels of academic promotion and leadership in Canada.

This is an interesting manuscript adressing gender representation of surgeons in academic leadership positions. I have the following questions

1. Were the academic positions in any way correlated with measurable output, e.g. publications ?

2. What do the authors believe is the reason for gender inequality in high posititions in the surgical field, the lack of access of women to training positions, lack of actual scientific output for various reasons (e.g. lack of funding, family obligations, lack of mentorship, etc.)

3. In the few cases in which women had leadership positions, was there a different situation with regards to promotion of women in their teams ?

Reviewer #3: The authors present an interesting manuscript on gender disparity among Canadian surgeons.

Please find my comments per section below:

Introduction:

Well done.

Methods:

- As correctly mentionned by the authors in the limitations section, publicly available website sources might have a potential for information bias. However, the approach seems reasonable to me as this source of potential bias is probably rather small.

- "Normally" age is adjusted for in most regression analysis. It would be interesting to see whether age plays a role. As a confounder or as an interaction term.

Results:

- It would be nice to have a "Table 1" displaying the raw data.

Discussion:

- The discussion is focussed on the situation in Canada (and sometwhat U.S.). It would be interesting so see some comparison with other countries discussed.

6. PLOS authors have the option to publish the peer review history of their article (what does this mean?). If published, this will include your full peer review and any attached files.

Reviewer #1: **Yes: **Eliane Angst

Reviewer #2: No

Reviewer #3: No

---

## [Author Response · Author response to Decision Letter 0]

30 Jul 2021

Academic Editor

From my side, I suggest to change the title: The term "review" may be misleading. In a review you describe what other authors have already published. Furthermore, reviews are not very welcome in PLOS ONE. Your study, however, is a real investigation. I suggest to change the title:.....a cross sectional investigation.

Response: Thank you for this suggestion.

Action Taken: We have changed the title to: 

“Gender imbalance amongst promotion and leadership in academic surgical programs in Canada: A cross-sectional investigation”

Response: We noted this request. We reviewed our reference list to ensure it is complete and correct. Regarding citations that have potentially been retracted, to the best of our knowledge, we are unaware of any of our citations that have been retracted. However, if the reviewers or editors have concerns that we may have missed or are not aware of, we are happy to make further edits to the list.

Action Taken: The reference list was reviewed, and in the process of double-checking, relevant links have been updated as necessary for applicable citations (#’s 3 and 4). Additionally, #16 has been updated to reflect the most current version of the Association American of Medical Colleges’ review of the status of women in academic medicine. 

Based on the Reviewer’s comments/suggestions, new references have been included in this manuscript. They are as follows: #’s 65 and 66 (see below). This also required re-numbering the reference list; the changes are reflected both in the reference list and the main text of the manuscript. 

65. Wolfert C, Rohde V, Mielke D, Hernandaz-Duran S. Female Neurosurgeons in Europe—On a Prevailing Glass Ceiling. World Neurosurgery. 2019;129: 460-466. doi:10.1016/j.wneu.2019.05.137

66. Wu B, Bhulani N, Jalal S, Din J, Khosa F. Gender Disparity in Leadership Positions of General Surgical Societies in North America, Europe, and Oceania. Cureus. 2019; 11(12): e6285. doi: 10.7759/cureus.6285

Reviewer #1

This study describes a cross-sectional online website review of the current faculty of surgical departments in Canada with the focus on gender distribution. The aim was to analyse the proportion of female in promotion and leadership. The data is put in relation to the politic of Canada, which tries to avoid gender disparity, as well as in context to the literature available to the topic and important networks like LinkedIn.

Response: Thank you for this positive summary.

The data do support an underrepresentation of women in surgery for promotion and leadership in general and relative to the number of trainees. The regression models may have been performed with very small numbers (n), leading to large confidence intervals. Therefore, it would be important to show a flow chart of the numbers assessed (initially and for each surgical specialty) and the numbers excluded (e.g. Obstetrics and Gynecology) and to add n to the figures (especially fig 4 to 7). With small n the regression may not be adequate statistically. 

Response: Thank you for this feedback. All three reviewers made some form of request for further information regarding our data, and we have interpreted this as a request to improve our overall data transparency. To clarify: the data collected for each surgeon was completed as stated in the methods and all of the collected data was included in the analysis, without exclusions. With respect to the regression models, thank you for your robustness in verifying the statistical analysis. Unsurprisingly, the general results have shown that there is an underrepresentation of women at the highest levels of promotion and leadership in surgical specialties across Canada. There are few women who have achieved these positions, leading to small absolute numbers. Despite the sample size, the regression analysis was able to additionally support our findings and show that gender continues to be a major driver of inequity at the highest levels of promotion and leadership even when accounting for other known factors. For this reason, we have included the regression in our paper as supporting evidence for our global findings. It is also in keeping with similar results from other works published in the field. 

Action Taken: To improve our data transparency, we have submitted Supporting information files S1 Tables 1a and 1b (academic surgical faculty) on Page 7 and S1 Tables 2a and 2b (resident surgeons) on Page 12 in the Results section. These are large datasets, so we have submitted the data organized by both specialty and by university. We felt as though, due to their size, these Tables were better suited as essential supporting files rather than as tables to feature in the Results section. Of course, we are happy to discuss this further with the PLOS ONE Editor(s).

As requested, figures 4-7 have been amended; x-axis labels were added, along with n-values to the y-axis. 

Also, we have added a statement about the statistical regression to our Limitations section on Page 16, noting the potential for small numbers (n) to impact the analysis:

“While the dataset is large in its entirety, at the highest levels of leadership and promotion, the absolute number of women remains low. In some instances, only a small n was available for the regression modelling and may impact the strength of the model. However, the models themselves have shown results in keeping with the global findings of the study, supporting the results along with those published by our peers with similar work in this field.”

Although the findings of the study are not new, it is impressive how we still underuse the potential of diversity: diverse teams get better results, which is essential in research and patient treatment. Moreover, we as a society cannot afford to train and develop females and not benefit from their knowledge. Therefore it is crucial to acknowledge how low the percentage of female is in position of promotion or leadership

Response: Thank you for these comments. We agree wholeheartedly.

The study conforms to the STROBE guidelines. 

The details are sufficient for the study to be reproduced

Response: Thank you for noting this.

The manuscript is well organized, but some improvement is suggested for better understanding:

Lines 300 to 303: make clear that women are 20% less likely to apply for a given job

Response: Thank you for your diligence. We agree that this phrasing could be improved, and we returned to the original source to ensure we had clarity on the correct phrasing for the reader.

Action Taken: Page 15, Lines 309-310 have been revised for clarity and understanding: 

“…thus, contributing to women applying to 20% fewer jobs than men.”

Label the x-axis in the figures 4-7, so it is self-explaining

Response: Thank you for pointing this out.

Action Taken: As noted above in our previous response, we have revised figures 4-7 by adding x-axis labels. 

Report all n

Response: We want to ensure that we are reporting our data according to the standards of PLOS ONE’s policy. As such, we have added this information into the manuscript. 

Action Taken: As noted above in our previous response, please refer to Supporting information files S1 Tables 1a and 1b (academic surgical faculty) on Page 7 and S1 Tables 2a and 2b (resident surgeons) on Page 12 in the Results section. These are large datasets, so we have submitted the data organized by both specialty and by university. We felt as though, due to their size, these Tables were better suited as essential supporting files rather than as tables to feature in the Results section. Of course, we are happy to discuss this further with the PLOS ONE Editor(s).

Show a flow chart and a chart with all the data according to PLOS ONE policy

Response: We note the requests for data transparency and want to ensure that we are submitting our data according to PLOS ONE Policies. All of the data we collected was used in the analysis and sub-analyses -- there was no data excluded.

Action Taken: See our responses above, and the submitted Tables 1a and 1b and Tables 2a and 2b with all of the data displayed. If there is further information that the reviewers and editors require, please let us know. 

Some abbreviations are not explained, please elaborate

Response: Thank you for bringing this to our attention.

Action Taken: The manuscript was reviewed throughout. Abbreviations were elaborated/changed to full-length text in multiple locations; please see the tracked changes for details.

Reviewer #2

The authors performed a cross-sectional online website review of current faculty listings for 17 university-affiliated academic surgical training departments across Canada in the 2019/2020 academic year. They conclude that women surgeons are significantly underrepresented at the highest levels of academic promotion and leadership in Canada. This is an interesting manuscript addressing gender representation of surgeons in academic leadership positions. 

Response: Thank you for this encouraging overview.

I have the following questions

1. Were the academic positions in any way correlated with measurable output, e.g. publications ?

Response: This is an interesting question. Unfortunately, the information required to answer this question is not available on publicly listed departmental websites and was not collected during our data-collection process to answer it fully within this investigation of all Canadian surgeons. However, we did note that academic positions for women were not correlated to attainment of a higher level of degree (Masters or PhD) despite proportionally more women than men having attained this level of additional education (page 6). In a similar study of Canadian General Surgeons, measurable outputs such as publications were not different between men and women (Reference #41, for interest).

Action Taken: None required.

2. What do the authors believe is the reason for gender inequality in high positions in the surgical field, the lack of access of women to training positions, lack of actual scientific output for various reasons (e.g. lack of funding, family obligations, lack of mentorship, etc.)

Response: Ongoing gender inequity in the surgical fields is a product of many intersecting issues. Research has shown increasing numbers of women in training and entering surgical careers over the last decades without similar increases in the number of women in leadership roles disproving the theory that this is purely a “pipeline” problem; we note this in the discussion on page 13. Instead, it is likely the ingrained culture of systemic biases that exist both in interpersonal work relationships and also in the metrics surrounding scientific achievement (grants, publications etc). These different aspects are reviewed in the discussion in some detail.

Action Taken: None required. As it currently stands, we feel as though we have postulated on the reason(s) as noted by Reviewer 2. If this is not sufficient, we would be happy to clarify further in the Discussion to highlight these multifaceted issues. However, to best enable this supplementary action, we are requested that Reviewer 2 please highlight specific changes/actions so we can tailor our response/edits accordingly. 

3. In the few cases in which women had leadership positions, was there a different situation with regards to promotion of women in their teams?

Response: Interesting question. The nature of our data collection would unfortunately not be able to determine if the method of promotion was different from one academic institution, or one surgical department, to another. However, as the number of women in leadership roles was so low, we did observe that there does not appear to be any trend related to surgical specialty or to academic institutions. 

Action Taken: None required.

Reviewer #3

The authors present an interesting manuscript on gender disparity among Canadian surgeons.

Response: Thank you for this affirmative synopsis.

Please find my comments per section below:

Introduction:

Well done.

Response: Thank-you.

Methods:

- As correctly mentioned by the authors in the limitations section, publicly available website sources might have a potential for information bias. However, the approach seems reasonable to me as this source of potential bias is probably rather small.

Response: Thank you for this comment. We acknowledge this limitation, but as you say, we feel that the potential bias is probably rather small.

- "Normally" age is adjusted for in most regression analysis. It would be interesting to see whether age plays a role. As a confounder or as an interaction term.

Response: Our variables of interest in the data set were categorical (man vs. woman; leadership position, yes or no; then what kind) and based on the data we could collect through the publicly available faculty websites. Age was unfortunately a variable that could not be reliably collected through our methodology. As such, we were unable to run a regression analysis with age as an interaction term.

Action Taken: None required.

Results:

- It would be nice to have a "Table 1" displaying the raw data.

Response: We want to ensure that we are reporting our data according to the standards of PLOS ONE’s policy as such we have added this information into the manuscript. 

Action Taken: Thank you Reviewer 3 for this feedback. This was noted by other reviewers as well. As such, please see our response to the previous reviewer. To improve our data transparency, we have supplied Supporting information files S1 Tables 1a and 1b (academic surgical faculty) on Page 7 and S1 Tables 2a and 2b (resident surgeons) on Page 12 in the Results section. These are large datasets, so we have submitted the data organized by both specialty and by university. We felt as though, due to their size, these Tables were better suited as essential supporting files rather than as tables to feature in the Results section. Of course, we are happy to discuss this further with the PLOS ONE Editor(s).

Discussion:

- The discussion is focused on the situation in Canada (and somewhat U.S.). It would be interesting to see some comparison with other countries discussed.

Response: Thank you for this comment. As our data reflects the current Canadian surgical environment, much of the discussion had reflected the comparative Canadian or North American surgical/medical academic literature. However, we agree with you that the world is becoming ever more connected and that the perspective of how this relates to the international surgical community is of interest and highly relevant. Much of our cited supporting literature reflects the status of gender in surgery and medicine from other nations beyond the US or Canada.

Action Taken: We have taken steps to highlight previously cited literature that reflects the international surgical community. Additionally, a literature review of gender and surgery was completed with a focus on international content. Relevant literature as it related to academia or leadership of women and women surgeons in medicine has been identified and the Discussion section on Page 14 was updated accordingly: 

“While our investigation highlights the current Canadian environment, the themes of ongoing gender disparity are not isolated to this country; with specific studies noting that inequities exist in Plastic Surgery, Neurosurgery and General surgery across North America, the EU and Oceania [39, 45, 46, 65, 66].” 

“It also mirrors work published by our international colleagues showing that women are underrepresented in medical leadership in the EU, Australia/Oceania and all over North America [17, 18, 66].”

---

## [Editor Report · Decision Letter 1]

16 Aug 2021

Gender imbalance amongst promotion and leadership in academic surgical programs in Canada: A cross-sectional investigation

PONE-D-21-18263R1

Dear Dr. Lefaivre,

We’re pleased to inform you that your manuscript has been judged scientifically suitable for publication and will be formally accepted for publication once it meets all outstanding technical requirements.

Kind regards,

Hans-Peter Simmen, M.D., Professor of Surgery

Academic Editor

PLOS ONE
---

## [Editor Report · Acceptance letter]

19 Aug 2021

PONE-D-21-18263R1 

Gender imbalance amongst promotion and leadership in academic surgical programs in Canada: A cross-sectional Investigation 

Dear Dr. Lefaivre:

I'm pleased to inform you that your manuscript has been deemed suitable for publication in PLOS ONE. Congratulations! Your manuscript is now with our production department. 

Kind regards, 

on behalf of

Dr. Hans-Peter Simmen 

Academic Editor

PLOS ONE